# The Effectiveness of Atezolizumab in Metastatic Large Cell Neuroendocrine Carcinoma of the Lungs: Insights from the LANCE Pilot Study

**DOI:** 10.3390/biomedicines12061161

**Published:** 2024-05-23

**Authors:** Georgios Evangelou, Ioannis P. Trontzas, Ioannis Gkiozos, Ioannis Vamvakaris, Christina Paraskeva, Maria Grammoustianou, Georgia Gomatou, Ioannis Tsamis, Ioannis Vathiotis, Maximillian Anagnostakis, Vasiliki Koliaraki, Kostas Syrigos

**Affiliations:** 13rd Department of Medicine, National and Kapodistrian University of Athens, Sotiria Chest Diseases Hospital, Mesogeion 152, 11527 Athens, Greece; john-tron@hotmail.com (I.P.T.); yiannisgk@hotmail.com (I.G.); marygrm90@gmail.com (M.G.); georgia.gomatou@gmail.com (G.G.); dr_tsamis_ioannis@hotmail.com (I.T.); johnvathiotis1@gmail.com (I.V.); maximilian2002@gmail.com (M.A.); ksyrigos@med.uoa.gr (K.S.); 2Department of Pathology, Sotiria Chest Diseases Hospital, Mesogeion 152, 11527 Athens, Greece; i.vamvakaris@yahoo.gr; 3Biomedical Sciences Research Center “Alexander Fleming”, Fleming 34, 16672 Vari, Greece; paraskeva@fleming.gr (C.P.);

**Keywords:** large cell neuroendocrine carcinoma, LCNEC, immunotherapy, atezolizumab

## Abstract

Background: Large cell neuroendocrine carcinoma (LCNEC) presents significant treatment challenges due to its rarity and limited therapeutic options. The LANCE study was designed to explore the survival benefits of incorporating atezolizumab in chemotherapy for metastatic LCNEC. Methods: In this non-randomized study, patients with metastatic LCNEC were prospectively enrolled and assigned to receive either standard chemotherapy plus atezolizumab followed by maintenance with atezolizumab or standard chemotherapy alone. The primary outcomes measured were 12- and 24-month survival rates, progression-free survival (PFS), and overall survival (OS) between the two groups. Results: Of the 22 patients screened, 17 met the inclusion criteria and received either atezolizumab plus platinum-based chemotherapy (*n* = 10) or chemotherapy alone (*n* = 7). After a median follow-up of 23.3 months, the 12-month survival rate was 57.1% (95% CI: 32.6–100%) and 14.3% (95% CI: 2.33–87.7%) for the atezolizumab and the chemotherapy-only groups, respectively. The survival benefit for the atezolizumab group was sustained at 24 months (45.7% vs. 14.3%). Overall survival was significantly higher for the atezolizumab group, and PFS was non-significantly associated with the addition of atezolizumab (log-rank *p* = 0.04 and 0.05, respectively). Conclusions: This pilot study suggests that the addition of atezolizumab to standard platinum-based chemotherapy may provide a substantial survival benefit compared with chemotherapy alone in the first-line treatment of metastatic LCNEC.

## 1. Introduction

Metastatic large cell neuroendocrine carcinoma (LCNEC) is a rare and aggressive type of cancer with an age-adjusted incidence rate of 0.3/100.000 [1]. Previously classified as a subtype of non-small cell lung cancer (NSCLC) by the World Health Organization, LCNEC was reclassified in 2015 to fall within the range of neuroendocrine neoplasms [2]. Unlike advances in treatments for other neuroendocrine neoplasms [3,4,5,6,7], such as neuroendocrine tumors (NETs) and small cell lung cancer (SCLC), therapeutic developments for metastatic LCNEC have been lagging and confined to chemotherapeutic drugs [8,9,10].

Because of the rarity of the disease, as well as its aggressiveness, which significantly hampers the investigation of new drugs, prospective evidence remains limited. The platinum–etoposide doublet has been considered the standard first-line regimen for metastatic LCNEC for over a decade based on the results of the phase 2 GFPC 0302 study [11]. More recently, the DART study investigated the efficacy of the nivolumab–ipilimumab combination in high-grade neuroendocrine carcinomas of the lung, among other neuroendocrine tumors. Besides low recruitment, this prospective study reported a significant response to immunotherapy in this subset of patients [12]. Additional evidence from retrospective studies suggests that a proportion of patients with LCNEC may benefit from immunotherapy with immune checkpoint inhibitors (ICIs) [13,14,15,16]. However, most of these studies used various combinations of ICIs with or without chemotherapy, thus failing to report a clear advantage of one drug over another or suggest a beneficial combination.

Atezolizumab (Tecentriq, F. Hoffmann-La Roche/Genentech), a humanized monoclonal antibody against programmed death ligand 1 (PD-L1), has shown efficacy in the treatment of metastatic SCLC and has subsequently changed the standard of care in the first-line setting of the disease based on the results of the IMPOWER 133 study [6]. Recent studies on the genetic and transcriptional aspects of LCNEC and SCLC indicate that these two diseases exhibit shared characteristics, including the neuroendocrine phenotype, high mutational burden, and aggressive biological behavior [17,18]. Considering the overlapping biological features and parallel chemotherapy strategies for both SCLC and LCNEC, investigation of the efficacy of atezolizumab in patients with LCNEC is deemed necessary.

## 2. Materials and Methods

### 2.1. Patient Selection and Pathology Review

In this single institution, non-randomized real-world study, we prospectively enrolled all consecutive patients with histologically confirmed metastatic lung-derived LCNEC who presented to the 3rd Department of Medicine of the National and Kapodistrian University of Athens, Sotiria Chest Diseases Hospital, between November 2019 and August 2022. Enrolled patients provided written informed consent. Patient enrollment fulfilled the following inclusion criteria: (i) histological diagnosis of lung-derived LCNEC by two independent pathologists, (ii) pathological or radiological confirmation of metastatic disease, and (iii) stable clinical condition (Eastern Cooperative Performance Status [ECOG PS] ≤ 2) to receive treatment upon physician evaluation. The second pathologist’s confirmation was sought after screening, and it was blinded to the initial report. In the event of a discrepancy, a third pathologist’s review was requested, and two unanimous reports were required for the final diagnosis. Exclusion criteria included (i) prior systemic treatment for their disease, (ii) ECOG PS > 2, and (iii) contraindication to receive treatment with ICIs. Patients with brain metastases upon initial diagnosis were considered eligible for enrollment after receiving whole-brain radiotherapy. All patients were monitored from the date of enrollment until August 2022. The study protocol was approved by the Institutional Review Board of Sotiria Chest Diseases Hospital (Athens, Greece; approval number: 4101/19) and was conducted in accordance with the Declaration of Helsinki. This study was registered at ClinicalTrials.gov (NCT06049966).

### 2.2. Study Design 

#### 2.2.1. Regulatory Engagement and Treatment Permissions

In this comparative cohort study, we evaluated the efficacy of atezolizumab combined with chemotherapy in patients with stage IV LCNEC, a patient group for which this regimen remains unapproved. Reflecting the complexities of real-world clinical practice, applications were submitted to the National Organization for the Provision of Health Services (EOPYY) for each patient to receive this innovative treatment, consisting of atezolizumab (1200 mg on day 1), carboplatin (AUC5 on Day 1), and etoposide (100 mg/m^2^ on Days 1–3) every 21 days for four cycles. Each application was independently assessed by different members of the EOPYY board.

Patients approved for this regimen were allocated to the experimental group, while those not approved constituted the control group, receiving only carboplatin and etoposide for up to six cycles. Following the initial treatment phase, patients in the experimental group who demonstrated a partial or complete response or stable disease as defined by RECIST 1.1 criteria continued with maintenance with atezolizumab every 21 days until disease progression or the occurrence of serious adverse events.

The primary reasons for treatment denial by EOPYY included significant comorbid conditions and performance status as determined by the Eastern Cooperative Oncology Group (ECOG) performance scale. Nonetheless, most rejections stemmed from stringent adherence to the regulatory guidelines concerning the use of atezolizumab.

In both arms, the patients continued with other treatments for disease progression based on the physician’s choice. Assessment of response to treatment with computerized tomography (CT) and physical evaluation was scheduled once upon completion of the combination treatment and then every three–four months unless a new onset of persistent symptoms or clinical deterioration of the patient indicated the need for early examination. The management of adverse effects and decisions regarding treatment discontinuation in patients receiving atezolizumab therapy followed the European Society of Medical Oncology (ESMO) guidelines for supportive and palliative care [17]. 

#### 2.2.2. Endpoints

The primary endpoints were overall survival (OS; time from treatment initiation to death from any cause) and progression-free survival (PFS; time from treatment initiation to disease progression according to RECIST or death from any cause, whichever occurred first). Patients with no events were censored on the date of their last follow-up. Censoring for OS and PFS was applied as of the last date on which the survival status was verified for patients who had not experienced documented disease progression and remained alive. Other key secondary endpoints were the overall response rate (ORR; the percentage of patients with a CR or PR according to RECIST).

#### 2.2.3. Sample Size 

This study, exploratory and designed to generate preliminary data on a rare condition, did not rely on traditional sample size calculations based on effect size estimation. Instead, the expected sample size of 20–25 participants was determined based on the historical recruitment rates for LCNEC studies and feasibility considerations, aiming to balance statistical needs with the practical challenges of studying rare diseases.

#### 2.2.4. Statistical Analysis

Statistical analysis was performed using R software (version 4.3.2). progression-free survival (PFS) and overall survival (OS) were estimated using the Kaplan–Meier method. To account for the differences in the times at which median PFS and OS were achieved across the various study arms, we conducted an analysis of survival probabilities at fixed time intervals of 12 and 24 months. We utilized the log-rank test to compare the survival distributions between groups and evaluate the efficacy of the treatments. Cox regression survival analysis was used to estimate adjusted hazard ratios and further evaluate the impact of treatment on survival outcomes. All statistical tests were two-sided, with the significance level set at α < 0.05.

#### 2.2.5. Data Availability

Detailed patient data and the R code used for statistical analyses are available at https://data.mendeley.com/datasets/64strcdfjh/1 (accessed on 21 February 2024) to ensure transparency and reproducibility of the findings. 

## 3. Results

### 3.1. Patient Screening and Enrollment

Between November 2019 and August 2022, 22 patients were prospectively screened for enrollment in the study. Of the 22 patients, 5 were excluded as ineligible because they had previously received chemotherapy and radiotherapy since they were initially diagnosed with limited-stage disease. The remaining 17 patients met all the eligibility criteria and were included in the study. The biopsies of the enrolled patients were reviewed by a second pathologist. In two patients, a third consultation was required to confirm the diagnosis of LCNEC. Of these patients, 10 were recruited to the experimental arm and treated with carboplatin, etoposide, and atezolizumab, while 7 patients were treated with carboplatin and etoposide and served as the control arm (Figure 1). The median follow-up duration was 23.3 months.

### 3.2. Baseline Characteristics

The baseline clinicopathological characteristics were well balanced between the two treatment groups (Table 1). The median age of patients in the experimental arm, receiving chemotherapy plus atezolizumab, was 68.5 years (range: 54–82), compared to 76 years (range: 53–85) in the control arm treated with chemotherapy alone. The majority of patients in both groups were male, accounting for 80% of the patients in the experimental arm and 85.7% in the control arm. Regarding smoking status, current smokers comprised 70% of the experimental group and 71.4% of the control group, with former and never-smokers accounting for a smaller proportion in both groups. All patients in both groups were Caucasians.

The time from diagnosis to treatment initiation was 35 days and 26 days in the experimental and control arms, respectively. The incidence of brain metastases was slightly higher in the control group (28.6%) than in the experimental group (20%). A similar pattern was observed for liver metastases (28.6% and 20% in the control and experimental groups, respectively). Regarding Eastern Cooperative Oncology Group (ECOG) Performance Status, both groups had similar distributions across categories 0, 1, and 2. TP53-positive immunohistochemistry (IHC) was observed in 80% of the experimental group and 71.4% of the control group. RB1-positive IHC was present in 40% of the experimental group and 42.9% of the control group. Previous studies have shown that mutations in TP53 were detected in 78% of LCNEC tumors, whereas the second most frequently mutated gene was RB1 (38%) [18]. The median Ki67% was 75% in the experimental arm and 70% in the control arm.

### 3.3. Effectiveness of Atezolizumab 

#### 3.3.1. Response Rate 

The objective response rate (ORR) was 50% for the experimental group (chemotherapy plus atezolizumab; ChT plus atezolizumab) and 42.9% for the control group (chemotherapy alone; ChT). At the conclusion of the study follow-up period, 40% of the individuals in the experimental group did not experience disease progression, in contrast to the control group, where none of the patients survived (Figure 2).

#### 3.3.2. Progression-Free Survival (PFS) 

We observed that the median progression-free survival (PFS) was not attained within the median follow-up period of 23.3 months in the ChT plus atezolizumab group, indicating prolonged periods without disease progression in a substantial proportion of patients. In contrast, the median PFS was 5.2 months (95% CI, 3.2–26.4) in the ChT group. In light of these findings, we present survival probabilities at fixed 12- and 24-month intervals to facilitate a more straightforward interpretation (Figure 3). These time points yielded crucial insights, with a 12-month estimated progression-free rate of 46.7% for the ChT plus atezolizumab group (95% CI, 23.3–93.6%) compared to 14.3% for the ChT group (95% CI, 2.33–87.7%). The 24-month rates mirrored these findings, underscoring the persistent trend. A log-rank test demonstrated a trend towards improved outcomes in the ChT plus atezolizumab group (*p* = 0.05). Additionally, the adjusted hazard ratio for PFS was 0.32 (95% CI, 0.10 to 1.04), implying a potential reduction in the risk of progression or death.

#### 3.3.3. Overall Survival (OS)

The median overall survival (OS) was not reached in the ChT plus atezolizumab group during the median follow-up of 23.3 months. In contrast, the median OS for the ChT group was 8.2 months (95% CI, 5.2–26.4). Given these findings, we presented survival data at the 12- and 24-month time points (Figure 4). This approach revealed a 12-month estimated survival rate of 57.1% for the ChT plus atezolizumab group (95% CI, 32.6–100%) versus 14.3% for the ChT group (95% CI, 2.33–87.7%). The 24-month survival rate for the ChT plus atezolizumab group was 45.7% (95% CI, 22.4–93.2%), demonstrating the sustained effectiveness of this treatment regimen. In contrast, the survival rate for the ChT group was maintained at 14.3% (95% CI, 2.33–87.7%), highlighting significant disparities in long-term outcomes between the treatment groups. The log-rank test indicated a significant difference in survival with a *p*-value of 0.04, and the hazard ratio for OS favored the ChT plus atezolizumab group (HR = 0.37, 95% CI, 0.10–1.05), suggesting a possible survival advantage.

### 3.4. Safety of Atezolizumab

Two patients who experienced grade 3 immunotherapy-related adverse events (AEs) were required to discontinue treatment and were administered prednisolone (1 mg/kg) until their symptoms improved to grade 1. Following the resolution of side effects, both patients were able to continue treatment with atezolizumab. The first patient developed a grade 3 skin reaction, specifically exfoliative dermatitis, on the dorsal and palmar surfaces of their upper limbs and on their trunk and thighs after the 10th cycle of maintenance treatment with atezolizumab. Lesions on the trunk and thighs had a less aggressive appearance than those on the upper limbs. Treatment with prednisolone resulted in the resolution of the lesions to grade 1, and the patient was able to continue treatment. However, by the end of the follow-up period, the toxicity remained at grade 1 but did not fully remit. The second patient experienced pneumonitis after six months of treatment with atezolizumab. Prednisolone treatment for two months resulted in the complete resolution of pneumonitis. No other Grade 3 or 4 immune-related events were observed in the study follow-up period.

### 3.5. Survival Analysis Using Cox Proportional Hazards Model

In a Cox proportional hazards regression analysis, the treatment variable showed an estimated adjusted hazard ratio (HR) of 0.33 (95% CI: 0.0177–6.33). The unadjusted HR for the treatment effect was 0.32 (95% CI, 0.10–1.05) *p* = 0.06) (Figure 5). 

In the multivariate analysis, the age of the patient was found to be a statistically significant factor impacting survival. However, liver or brain metastasis, sex, RB1 and TP53 mutations, and Ki67 levels were not determined to have a statistically significant impact on survival.

“First response” was defined as response or not based on the RECIST criteria after completion of four cycles of treatment and was characterized as either (1) complete response (CR), partial response (PR), stable disease (SD), or (2) disease progression (PD). The first response to treatment was a strong predictor of better survival, with a 98% reduction in the hazard ratio (HR, 0.02; 95% confidence interval [CI], 0.00085–0.48; *p* = 0.016), indicating that patients who responded (CR/PR/SD) during the first four cycles of treatment had significantly improved survival rates. The model demonstrated good predictive ability with a concordance index of 0.89 and a global *p*-value of 0.002 from the log-rank test.

## 4. Discussion

The investigation of new drugs in rare cancers, such as LCNEC, carries several difficulties, which ultimately result in limited and protracted approvals. Despite the advancement in the management of several neuroendocrine tumors, including the introduction of ICIs in the management of NETs and SCLC [4,6,19,20], treatment options for metastatic LCNEC patients remain limited and mostly confined to chemotherapy regimens [21]. We investigated the effectiveness of atezolizumab in addition to standard chemotherapy in patients with metastatic LCNEC.

To the best of our knowledge, the LANCE study is the first prospective real-world study to test this regimen for LCNEC. The allocation of patients into an experimental group, which consisted of atezolizumab in combination with chemotherapy followed by atezolizumab alone as maintenance therapy, and a control group that received chemotherapy alone, provided an opportunity to directly evaluate the effect of adding atezolizumab on response and survival outcomes. The median PFS and OS were not reached for the atezolizumab group, while those for the control group were 5.2 months (95% CI, 3.2–26.4) and 8.2 months (95% CI, 5.2–26.4), respectively. The benefit for the experimental arm was not significant for PFS (log-rank *p* = 0.05, HR = 0.32, 95% CI: 0.10 to 1.04) and for OS (log-rank *p* = 0.04, HR = 0.37, 95% CI: 0.10 to 1.05); however, considering the early and sustained dissociation of the survival curves of the two groups, we believe that the impact of the treatment was hampered by the small sample size. Projected progression-free and survival rates at 12 and 24 months were calculated to interpret the magnitude of the intervention’s impact more accurately, which demonstrated a clear benefit for both progression-free and survival rates at 12 months in the atezolizumab group (46.7% vs. 14.3% and 57.1% vs. 14.3%, respectively). The benefit for the experimental arm was maintained at the 24-month projected rate. An additional surrogate to atezolizumab’s benefit was the number of patients who were alive at the end of the follow-up period, with 40% of the patients in the experimental arm being alive without evidence of disease progression compared with no survivor in the control arm.

In addition, we conducted a multivariate Cox regression analysis to examine the influence of various clinicopathological parameters, including age, sex, de novo liver and brain metastases, and immunohistochemical markers TP53, RB1, and KI67, on patient outcomes. Our analysis revealed that older age is significantly associated with poorer survival rates. Furthermore, we observed that “first response”, which was evaluated after the induction phase of the regimen according to RECIST, appeared to be a strong predictor of patient outcomes. Further studies are necessary to confirm these findings.

Regarding the safety profile of atezolizumab, no unexpected or unacceptable irAEs or other adverse events were documented. All toxicities were managed adequately according to ESMO guidelines, and no treatment discontinuation was observed in either group. Two patients with grade 3 irAEs (skin toxicity and pneumonitis) withheld atezolizumab therapy and received high-dose corticosteroids until resolution of symptoms to grade 1 when they continued therapy. These findings are consistent with the anticipated atezolizumab irAEs reported in other studies [22,23,24].

The present study has several limitations. First, a prespecified power sample analysis was not sought, and enrollment was relatively small. However, given the rarity of the disease, the 17 patients enrolled in this study represent one of the largest series of immunotherapy studies on LCNEC, highlighting the challenges of conducting large-scale studies on rare diseases. Second, this single-center study enrolled patients from the same demographic group, which may hamper the reproducibility of our findings in different populations. An additional pitfall was the non-randomized design, which was implied by the necessary compliance with regulatory treatment permissions by national authorities. This limitation highlights real-world complexities in the management of patients with rare malignancies. Although these limitations are acknowledged, they can be addressed in future large multicenter studies. 

Overall, this pilot study demonstrated prospective data supporting the survival benefit of atezolizumab addition in patients with metastatic LCNEC receiving platinum-based chemotherapy. Our findings provide further evidence for the use of immunotherapy in LCNEC and underline the necessity for further research on new drugs for this rare disease.

## 5. Conclusions

The conclusions drawn from the LANCE study underscore the potential of atezolizumab in combination with chemotherapy as a promising therapeutic avenue for patients suffering from metastatic large cell neuroendocrine carcinoma of the lung. Our study, albeit pilot in nature, reveals a notable improvement in progression-free survival and overall survival rates among patients treated with combination therapy compared to those receiving chemotherapy alone. This indicates not only the therapeutic efficacy of atezolizumab when paired with chemotherapy but also highlights the necessity for further exploration into the integration of immunotherapy within standard treatment protocols for metastatic LCNEC.

## Figures and Tables

**Figure 1 biomedicines-12-01161-f001:**
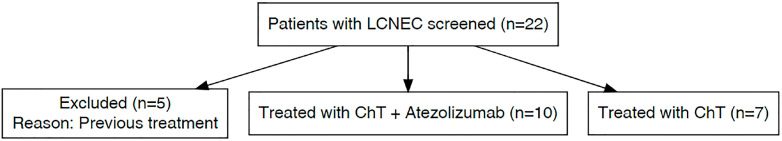
Patient flowchart and time sampling, LCNEC: large cell neuroendocrine carcinoma, ChT: chemotherapy.

**Figure 2 biomedicines-12-01161-f002:**
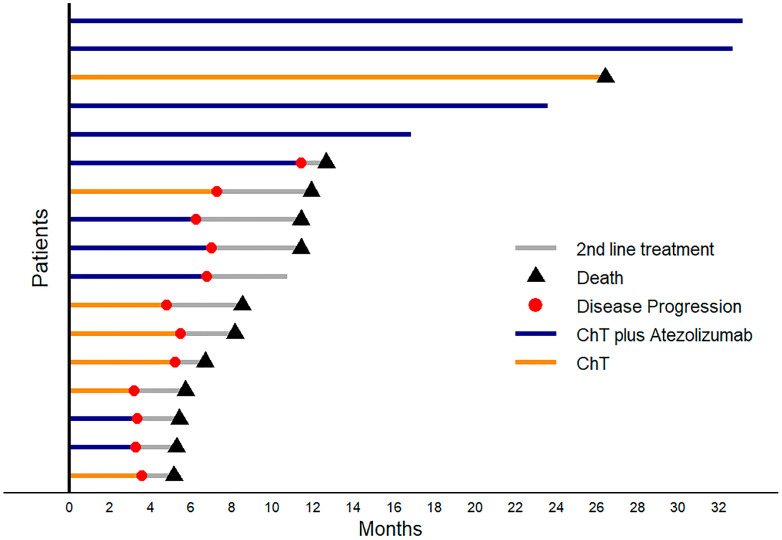
Swimmer’s plot of patient overall survival showing the type of treatment, point of disease progression, and death for each patient and the patients who received 2nd line treatment.

**Figure 3 biomedicines-12-01161-f003:**
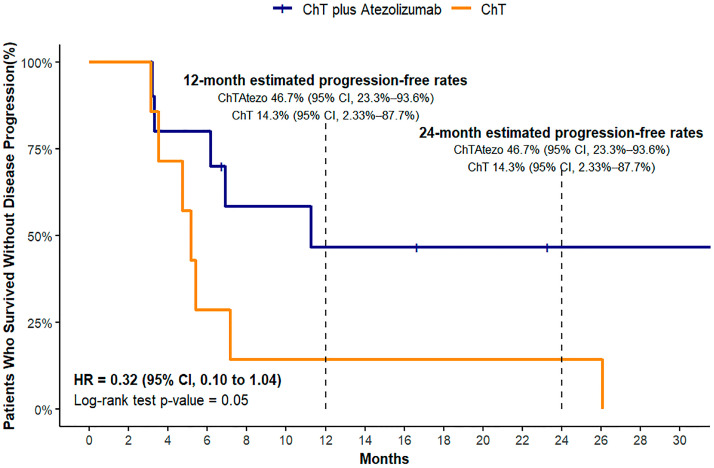
Kaplan–Meier estimates for progression-free survival.

**Figure 4 biomedicines-12-01161-f004:**
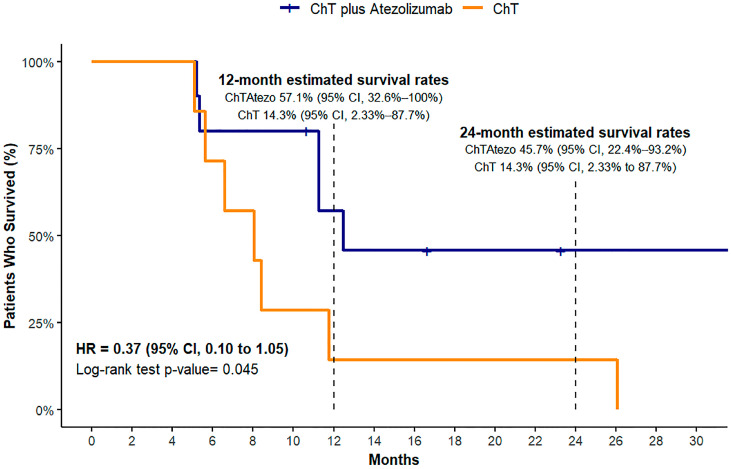
Kaplan–Meier estimates for overall survival.

**Figure 5 biomedicines-12-01161-f005:**
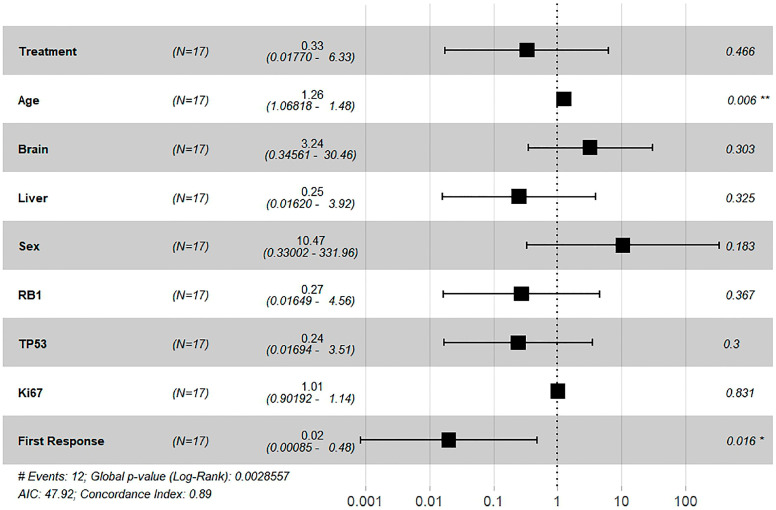
Cox regression survival analysis, including the treatment, age, sex, de novo brain and liver metastasis, ki67, TP53, RB1 status, and first response. * *p* < 0.05; ** *p* < 0.01.

**Table 1 biomedicines-12-01161-t001:** Patient characteristics. Abbreviations: IHC = immunohistochemistry, ECOG = Eastern Cooperative Oncology Group.

	Chemotherapy Plus Atezolizumab (%)	Chemotherapy (%)	*p* (*t* Test)
Age	68.5 (54–82)	76 (53–85)	0.31
Male sex	80	85.7	0.79
Smoking Status			0.80
Current	70	71.4	
Former	20	14.3	
Never	10	14.3	
Race Caucasian	100	100	
Time from diagnosis to treatment (days)	35	26	0.77
Brain metastases	20	28.6	0.74
Liver metastases	20	28.6	0.71
ECOG Performance Status (%)			0.48
0	30	28.6	
1	40	42.8	
2	30	28.6	
TP53 positive IHC	80	71.4	0.71
RB1 positive IHC	40	42.9	0.91
Ki67% (median)	75	70	0.93

## Data Availability

The dataset derived from the LANCE study comprising patient outcomes, such as overall and progression-free survival, demographic information (age and gender), performance status, treatment regimens, and immunohistochemical markers (TP53, RB1, and Ki67%), has been deposited in https://data.mendeley.com/datasets/64strcdfjh/1 (accessed on 21 February 2024).

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
