# Peer review of "The Effectiveness of Atezolizumab in Metastatic Large Cell Neuroendocrine Carcinoma of the Lungs: Insights from the LANCE Pilot Study"

_biomedicines, 2024, doi:10.3390/biomedicines12061161_

Round 1

Reviewer 1 Report

Comments and Suggestions for Authors

In this manuscript, the author demonstrates the potential benefits of incorporating atezolizumab into standard chemotherapy for treating patients with LCNEC. The methodology outlined in the main text and supplementary materials is presented clearly, although the language could benefit from refinement by a native speaker.

Major comments:

The authors should elucidate the rationale for analyzing miR-375 in the study. Additionally, the use of miRNA as a biomarker is uncommon in clinical practice, suggesting that investigating the expression of many cytokines and proteins might yield more meaningful insights. If the authors choose to analyze miRNAs, they should consider screening other miRNA expressions, particularly comparing their expression levels in patients' tumor samples to those from healthy donors. Moreover, is there a correlation between miR-375 and the efficacy of atezolizumab? Given its established role as a checkpoint inhibitor, further exploration of its effects and mechanisms on immune cells would be beneficial.

Furthermore, the Discussion section appears overly lengthy and repetitive, warranting condensation. Additionally, the potential translational implications of the study should be elaborated upon.

Comments on the Quality of English Language

Minor edition needed.

Author Response

Thank you for your insightful comments and suggestions.

After careful consideration of your feedback, as well as the comments from another reviewer and the academic editor's notes, we removed the analysis of miR-375 from the main text of our manuscript. This decision reflects a comprehensive review of all the guidance provided, ensuring that our study maintains a focused and clinically relevant approach.

The rationale for including miR-375 analysis was originally supported by several prior studies, indicating its relevance in neuroendocrine neoplasms (Korotaeva et al., 2021; Detassis et al., 2020; Nanayakkara et al., 2020; Zatelli et al., 2017). Collectively, these studies suggest that miR-375 and other miRNAs can play a significant role in distinguishing and characterizing various tumor types, including their prognostic capacities.

However, we acknowledge the reviewer’s concern regarding the limited use of miRNAs as biomarkers in clinical practice and the marginal correlation found in our study between miR-375 and disease progression. This led us to conclude that its presence in the main manuscript might not be justified, aligning with the suggestion of exploring the expression of a broader array of cytokines and proteins, which should be considered in future studies.

By excluding the miR-375 data from the main manuscript, we have also condensed the Discussion section, thus enhancing the overall readability and coherence of the text as recommended.

Lastly, in response to the feedback on refining the language for clarity and fluency, we sought the assistance of a native English speaker to improve the language quality of the manuscript.

We appreciate the reviewer's constructive feedback, which has significantly helped to refine our manuscript. We hope that these revisions address the concerns raised and make this study more robust and relevant to the field.

Thank you again for your thoughtful review and guidance.

Reviewer 2 Report

Comments and Suggestions for Authors

This is a nice study devoted to a rare category of lung cancer patients. There are several major issues which need to be addressed: 1) This is small hypothesis-generating study. Please try to shorten paper at leas twice by deleting unnecessary information or transferring some text, table and figures to the Supplement. 2) Significant scientific/English editing is required: many fragments of the text are self-explanatory, do not need to be so extensive; 3) Why the authors decided to measure miR-375? This is not explained in the Introduction; 4) What was the treatment after the progression? 5) There are some frank mistakes, for example “chemotherapy-atezolizumab conjugate” (line 369). Conjugate and combination are different things!   

Comments on the Quality of English Language

Moderate scientific/English editing is required. 

Author Response

Thank you for your constructive comments and suggestions on our manuscript. We have carefully considered each point to enhance the clarity and scientific rigor of our study regarding this rare category of lung cancer. Below, we describe the revisions made in response to the issues raised.

  1. Manuscript Length and Content: Following your advice and in agreement with other reviews, we have excluded miR-375 data from the main text. These data showed a non-significant marginal correlation with disease progression and were excluded. We have also removed extensive, self-explanatory text, contributing to a more concise manuscript.
  2. Scientific and English Editing: We thoroughly revised the manuscript for both content and language clarity. Redundant fragments were eliminated, and the text was refined to improve readability and scientific communication.
  3. Rationale for miR-375 Analysis: The decision was based on previous studies indicating its potential role in neuroendocrine neoplasms, such as those by Korotaeva et al. (2021), Detassis et al. (2020), Nanayakkara et al. (2020), and Zatelli et al. (2017), which suggests that miR-375 and other miRNAs are significant in distinguishing and characterizing tumor types. In response to feedback concerning the inclusion of miR-375 data in our study, we excluded this data from the manuscript. The miR-375 data did not show a significant correlation with disease progression or treatment response in the context of atezolizumab and chemotherapy, and its inclusion could potentially detract from primary findings, which are of greater clinical relevance to our readership. By removing these data, we aim to enhance the clarity and focus of our manuscript on the more impactful results regarding the effectiveness of atezolizumab in treating metastatic large-cell neuroendocrine carcinoma. This approach aligns with editorial guidance to prioritize findings that provide clear insights and meaningful contributions to the field. Recognizing the interest in miRNA analysis within cancer research, we acknowledge that the findings related to miR-375 should be better explored in a more detailed study focused on miRNA profiling. We plan to pursue this in future studies and consider incorporating a broader range of miRNA and cytokine analyses to enrich our understanding of their roles in such cancers.
  4. Treatment Post-Progression: As mentioned, the treatment options following disease progression were based on the physician's discretion, which is now explicitly noted in the manuscript. We confirm that none of the patients received immune checkpoint inhibitors in subsequent lines of therapy, and those who received second-line treatments were administered alternative chemotherapy agents.
  5. Correction of Errors: We corrected the grammatical and syntax errors identified throughout the manuscript.

We appreciate the opportunity to refine our work, and believe that these revisions have significantly improved the manuscript. We are grateful for your thorough review and look forward to your feedback regarding the updated version.

Thank you once again for your valuable suggestion.

Round 2

Reviewer 2 Report

Comments and Suggestions for Authors

-

Comments on the Quality of English Language

Minor editing is required